# On Developing a Hydrophobic Rubberized Cement Paste

**DOI:** 10.3390/ma14133687

**Published:** 2021-07-01

**Authors:** Chi-Yao Chen, Zih-Yao Shen, Maw-Tien Lee

**Affiliations:** Department of Applied Chemistry, National Chiayi University, Chiayi City 600355, Taiwan; justin@shinchuang.com (C.-Y.C.); mark@shinchuang.com (Z.-Y.S.)

**Keywords:** partial oxidation, crumb rubber, FT-IR, AFM, SEM, XRD, NMR

## Abstract

It is well known that most cement matrix materials are hydrophilic. For structural materials, hydrophilicity is harmful because the absorption of water will induce serious damage to these materials. In this study, crumb rubber was pretreated by partial oxidation and used as an additive to develop a hydrophobic rubberized cement paste. The pretreated crumb rubber was investigated using Fourier-transform infrared spectrometry (FT-IR) to understand the function groups on its surface. The pyrolysis oil adsorbed on the surface of the crumb rubber was observed by FT-IR and nuclear magnetic resonance (NMR) spectroscopy. A colloid probe with calcium silicate hydrate (C–S–H) at the apex was prepared to measure the intermolecular interaction forces between the crumb rubber and the C-S-H using an atomic force microscope (AFM). Pure cement paste, cement paste with the as-received crumb rubber, and cement paste with pretreated crumb rubber were prepared for comparison. FT-IR, X-ray diffraction (XRD), and scanning electron microscopy (SEM) were used to understand the microstructure of the pastes. The static contact angle was used as the index of the hydrophobicity of the pastes. Experimental results showed that the hardened cement paste containing partially oxidized crumb rubber had excellent hydrophobic properties with an insignificant reduction in the compressive strength.

## 1. Introduction

Cement–matrix composites, including concrete (with fine and coarse aggregates), mortar (with fine aggregate), and paste, are widely used in building materials [1]. A cement–matrix composite with the combination of two or more non-miscible phases results in a macroscopically non-homogeneous material. In comparison with cement paste alone, the additives will affect the hydration of the cement and change the properties of the matrix in terms of strength, weight, stiffness, cracking resistance, etc. Most of the properties cannot simply be deduced from the rule of mixture, but should be identified by analyzing the physical interaction through their interfaces [2]. Therefore, many studies have been conducted to explore the interaction between a matrix and additives to improve the physical properties of the composite. There are many additives that have the function of enhancing the structural performance of a composite. These additives can improve the properties of the composites, such as strength, stiffness, fatigue, toughness, etc. However, overcoming hydrophilicity is still a problem.

Thomas et al. [3] reviewed admixtures in cement–matrix composites and suggested that introducing advanced new properties is critical for developing new cement–matrix composites. In the context of additives, an increasingly important option is materials with a low environmental impact. Such materials can be recycled additives whose disposal would cause a negative environmental impact, or manufactured materials that possess cementitious characteristics produced at a lower environmental cost. Waste tire rubber is one of the possible alternatives.

The disposal of waste tires is an issue of growing concern worldwide. The ever-increasing volume of rubber waste in landfills from the disposal of used tires has grown into a serious environmental problem. A large amount of waste tires are generated worldwide every year. As waste tires are not easily biodegradable after a long period of landfill, this is not only an environmental problem but also a waste of natural resources. Many researchers [4,5,6] have suggested promising alternatives such as (1) mixed rubber, asphalt, and cement; (2) fuel; and (3) reuse. It is well known that crumb rubber can be used as an additive to form a cement–matrix composite known as rubcrete [6,7,8,9,10,11,12,13,14,15,16,17,18,19,20,21]. Some excellent review papers gave detailed discussions on rubberized concrete [22,23,24]. However, hydrophobic crumb rubber does not distribute well in the cement hydrate, and it cannot bond firmly with the cement hydrate. To improve the mechanical strength of rubcrete, many studies have tried to modify the surface properties of the rubber particles to enhance their adhesion to C–S–H. For example, Segre and Joekes [25] and Chou et al. [26] used NaOH to modify the surface of waste tires. Lee et al. [27] treated rubber with HNO_3_ and a METHOCEL cellulose ether solution. Li et al. [28] pre-coated a cement paste on rubber particles. However, none of the surface treatments mentioned above have shown significant improvements in preventing the huge strength loss due to the incorporation of rubber particles/chips.

Huang and his research team developed a two-stage surface treatment to improve the properties of rubber-modified cement composites and concrete [29,30,31]. They used a silane coupling agent to develop chemical bonds between rubber particles and cement. Furthermore, by developing a hard shell around the rubber particles with cementitious materials, they successfully increased the stiffness of the rubber particles.

Chou et al. [32] used organic sulfur compounds to enhance the interaction force between rubber and cement hydrates. They further improved the rubberized mortar by partially oxidizing the crumb rubber to enhance its hydrophilic property [33]. In their study, the partially oxidized crumb rubber was investigated with Fourier-transform infrared spectroscopy (FT-IR) and they found the existence of hydrophilic functional groups of sulfoxide on the surfaces of the treated crumb rubber.

Wua et al. [34] used pure C_3_S to replace cement to react with pure water and found that with the addition of partially oxidized rubber, the C-S-H red-shifted in the FT-IR spectrum in comparison to the C-S-H from the pure C_3_S. Liu et al. [35] observed the effect of octyl phenyl sulfoxide on cement hydration. They concluded that the addition of organic sulfoxide can slow the hydration rate and bring about better crystallization. Therefore, the long-term microstructures of hardened cement paste with organic sulfoxide are denser than those of pure cement hydrates.

It is well known that the hydrophilicity of cement–matrix composites is detrimental for their use as structural materials, because they will absorb moisture and cause serious damage. Therefore, the outer surfaces of most of buildings are treated with hydrophobic materials, such as hydrophobic organic paints, to reduce the absorption of water. Yen et al. [36] developed a rubberized paste with water-repellant properties. However, they did not give the mechanical properties of the paste.

In this study, a new rubber cement paste was developed by using partially oxidized crumb rubber as additive. This rubberized cement paste had bulk hydrophobic properties, and the compressive strength was not significantly reduced as compared with that of pure cement paste.

## 2. Materials and Methods

### 2.1. Materials

The rubber (styrene butadiene rubber (SBR) + natural rubber) used in this study was supplied by Chih Cheng Rubber Factory Co. LTD, Chia-Yi, Taiwan. The crumb rubber was screened to 300–600 μm for use as an additive of cement–matrix composite. The rubber sheet (10 × 10 × 0.1 mm^3^) was polished and partially oxidized. The surface morphology of the rubber sheet was scanned by AFM to understand the effect of partial oxidation on the surface morphology. A colloidal probe having a C-S-H tip was prepared for measuring the intermolecular interaction force between rubber and C-S-H. The first type of Portland cement was used in this study.

### 2.2. Experimental Procedure

The experiment was carried out in a furnace (155 × 300 × 155 mm^3^) with controllable temperature. The rubber sheet was suspended in a furnace with a wire, and the crumb rubber was placed in a cylindrical container having a diameter of 80 mm and a height of 50 mm. Thermocouples were used to measure the temperature in the center of the container (T1), above the crumb rubber (T2), and near the container (T3 and T4) for further analysis. The experimental apparatus and the layouts of the thermocouples are depicted in Figure 1. The furnace temperature was set at 320 °C. After the power was turned off, the furnace was left overnight to cool under ambient conditions.

The original and the treated rubber sheets were scanned with an AFM (Axiovert 200) to obtain their surface morphology. Prior to scanning, the rubber sheets were cleaned with acetone and distilled water to remove possible contaminants from the surface. To measure the intermolecular interaction force, a colloidal probe having C-S-H at its tip was prepared according to the method proposed by Plassard et al. [37]. Chou et al. [32] successfully applied this colloidal probe to measure the intermolecular interaction between C-S-H and additives. Rubberized cement pastes were prepared with 5% crumb rubber. Both the as-received and the treated crumb rubber were used for comparison. The microstructure of the paste was observed by XRD (Shimadzu XRD-6000), AFM (JPK, Axiovert 200), and SEM (Hitachi SU8010). FT-IR (Shimadzu FTIR-8400) was used to explore the variation of the function groups. The scanning region was from 4600 to 400 cm^−1^, with a resolution of 8.0 cm^−1^. The transmittance mode was used. Crumb rubber and KBr were mixed and pressed to be a thin film.

During the partial oxidation of the rubber, a small part of the rubber was cracked, and the generated pyrolysis oil was adsorbed on the rubber surface. In order to understand the characteristics of the pyrolysis oil and the possible influence on the subsequent cement paste, water and acetone were used to extract it separately. The extraction procedures were as shown in Figure 2a.

The absorbed pyrolysis oil extracted with acetone contained less polar or non-polar compounds (Figure 2b). They were first extracted with acetone and then concentrated under a reduced pressure concentrator and finally purified by CDCl_3_. The products (Figure 2c) extracted with deionized water and purified by D_2_O were polar compounds. Both samples were analyzed by NMR (Agilent Technologies NMR 400 Hz) with a scan rate of 5000 times in 10 h. Functional groups in the carbon chain were presumed based on the 1H-NMR and ^13^C-NMR spectra.

The static contact angle (Sindatek Model 100SB) was measured as the hydrophobicity index. Five microliters of water were placed on the surface of the hardened paste with a micropipette, and the static contact angle was measured using an image of the water droplets on the surface. The compressive strength of the specimen was measured according to the American Society for Testing and Materials (ASTM) C109. For statistical evaluation, 9 samples were made for each experiment. The treated and untreated rubber is added for 5 wt% of cement. The compressive specimen (50 × 50 × 50 mm^3^) was prepared with a w/c value of 0.35. There were six specimens for each curing age. The curing process was carried out in saturated lime water, and the temperature was controlled at 23 ± 2 °C.

## 3. Results

### 3.1. Rubber Particles Analysis

#### 3.1.1. Temperature Change inside and around the Reactor

In this study, the temperature of the furnace was set at 320 °C. The temperatures of the four points were measured and recorded as shown in Figure 3. The two temperatures near the container and the temperature above the rubber rose linearly as the heating time increased. However, the temperature in the middle of the rubber particles rose slowly compared to the temperature at the other three points. It is due to the low thermal conductivity of the crumb rubber. The power was turned off after heating for 50 min. Even when the power was turned off, the temperature at the center of the granular rubber continuously increased to approximately 400 °C. This was due to the pyrolysis reaction occurring following the oxidation reaction. It is well known that most oxidation reactions are exothermic reactions. Table 1 shows the energy required to cleave the C-S and C-C bonds. It is clear that the S-S, Sx, and C-S bonds require less energy to be cut than the C-C bond does [38,39,40,41]. As a small amount of oxygen was present in the reactor system in the initial stage, the crumb rubber was oxidized to produce new chemical bonds, such as S = O, O-S-O, and C-O, and emitted heat. The temperature was raised. However, the furnace was oxygen deficient, as when oxygen was consumed, the oxidation reaction ceased. At the same time, the high temperature caused an endothermic pyrolysis reaction and the temperature was lowered [42,43]. When the temperature drops to a certain low level, the cleavage reaction will stop.

#### 3.1.2. FT-IR Analysis for Crumb Rubber and Pyrolysis Oil

Figure 4 shows the IR spectrum of the as-received crumb rubber, partially oxidized crumb rubber, and pyrolysis oil. Here, A represents the spectrum of the original crumb rubber—this is for comparison; B is a partially oxidized crumb rubber; and C is the pyrolysis oil. It is clear that the treated crumb rubber and the pyrolysis oil have more hydrophilic S-Ox bonds than the as-received crumb rubber does. This is similar to the results of Yang and Lee [44]. These polar function groups can enhance the hydration of cement and lead to an increase in the mechanical properties of cement paste [35].

With the facts discussed above, it is suggested that the possible mechanism of the partial oxidation and the pyrolysis reaction of the crumb rubber can be depicted as in Figure 5. The pyrolysis oil is a very complex mixture with various S-O bonds. In our experiments, the light fraction of the pyrolysis gas was adsorbed by a mixture of soil and sand. The heavy portion was adsorbed on the surface of the crumb rubber or condensed on the reactor wall.

#### 3.1.3. NMR Analysis of Absorbing Pyrolysis Oil

In order to understand the pyrolysis oil adsorbed on the crumb rubber, the oil was analyzed using NMR.

Figure 6a shows the ^1^H NMR spectrum of the component in Figure 2b. The signals in Figure 6a are first-order carbon (0.8~1.0 ppm), secondary carbon (1.0~1.4 ppm), and third-order carbon (1.6~1.8 ppm) [45], and the R-SH signal is between 1.5 and 1.6 ppm [44,46]. The sulfur compound Rx-SOz-RyH has signals at 2.2~2.4 ppm [46,47,48]. R-CH_2_-SO_2_X and CX-SOxH appear at 3.6 and 5.2 ppm [46,48], but the signals of the above products are less obvious. The ^13^C NMR spectrum (Figure 6b) shows that the signal is in the range of 10–30 ppm, which is similar to the reference result [46]. This information is not sufficient to prove the existence of polar functional groups in the absorbed pyrolysis oil. As shown in Figure 2b, the polar components of pyrolysis oil are difficult to extract using acetone. Therefore, the amount of polar functional groups is lower than the detection limit.

To solve this problem, water was used to extract the adsorbed pyrolysis oil, as shown in Figure 2a. The concentrated components are shown in Figure 2c. Figure 6c,d are the ^1^H-NMR and ^13^C NMR spectra of the components in the concentrated samples (Figure 2c), respectively. In Figure 6c, the signals between 2 and 4 ppm are sulfur oxides. In Figure 6d, the signals between 40 and 80 ppm are the carbon adjacent to sulfur oxides [46]. The polar sulfur oxides connect to 1° carbon, 2° carbon, and 3° carbon. As parts of the sulfurized products are connected to the higher-order carbon, many signals are less obvious. However, ^13^C NMR provides evidence that the pyrolysis oil has polar function groups.

### 3.2. Rubber Specimens Analysis

#### 3.2.1. Surface Morphology

The surface morphology of the rubber sheets is shown in Figure 7a,b. Figure 7a shows the as-received rubber, which had a smooth surface. The surface of the treated rubber is shown in Figure 7b, which was rippled and rough. The increase in surface ripple enhances the interaction between the crumb rubber and cement hydrate [25,49,50]. The compressive strength of cement paste containing treated rubber is, therefore, greater than that of cement paste containing untreated rubber [4,26,37].

#### 3.2.2. Intermolecular Forces

The intermolecular interaction forces between the C–S–H colloid probe [32] and the rubber were measured with an AFM. The force distribution of the treated rubber is presented in Figure 8. The results show that for the untreated rubber (UR), the intermolecular force was 100~300 nN, and the intermolecular force for treated rubber (TR) was 400~700 nN. The treated rubber clearly provided larger intermolecular interaction forces than those of the as-received rubber. This is due to the production of polar function groups on the rubber surface.

### 3.3. Cement Paste

#### 3.3.1. FT-IR Analysis

The cement pastes were observed with FT-IR to understand the microstructures of the pastes. The FT-IR absorption spectra of hardened pastes at 28 days are shown in Figure 9. There are three spectra in this figure: the paste with 5%, by weight, of the as-received crumb rubber (UR5); the paste with the 5%, by weight, treated crumb rubber (TR5); and the pure cement paste (C0). The bands representing various functional groups are depicted in Table 2 [51,52,53,54,55]. The absorption within 800~1200 cm^−1^ is due to functional groups, Si-O and S = O, etc. The absorption near 900 cm^−1^ is C-S-H [3,7,33,34].

Figure 9 shows the splitting of the δO-H absorption signal of TR5 water (band a, 1666 cm^−1^, 1635 cm^−1^). It is different from that of C0 and UR5. This is due to the interaction of the rubber particles with the RSOx functional group produced during partial oxidation (Figure 4). The RSOx functional group is polar, which can enhance the film flow of water in the cement slurry [26] and improve the cement hydration reaction. Positions 9 and h are the vibration modes of Si-O. It is clear that the absorption at position h (the as-received rubber) splits into two peaks (h and i), while the absorption at position 9 (the treated rubber) does not split. Since the as-received rubber is hydrophobic, the water in the paste cannot be evenly distributed in the cement and rubber, and the signal is more easily divided into two and recognized by the instrument [56].

#### 3.3.2. XRD Analysis

Figure 10 shows the XRD patterns of the C0, UR5, and TR5 pastes. The position 2θ = 29.4° is C-S-H crystal. According to the Scherrer equation (Equation (1)), the particle size (τ) of the crystal is inverse to the FWHM (full width at half maximum β). The values of FWHMs of C0, TR5 and UR5, are shown in Table 3. Figure 10 and Table 3 show that the FWHM of TR5 is the largest, and that of UR5 is the smallest. This proves that the addition of rubber affects the hydration of cement. The as-received rubber is hydrophobic, which may retard the transportation of water within the paste and affect the hydration of cement. However, the partially oxidized rubber has hydrophilic functional groups on its surface that will enhance the film flow of water within the paste and contribute to the hydration of cement. In addition, the hydrophilic surface facilitates the adhesion of the hydrate and rubber.
(1)$$\tau =\frac{K\lambda }{\beta \cos\theta }$$

#### 3.3.3. SEM Analysis

The microstructures of C0, UR5, and TR5 are shown in Figure 11a–c. The cement paste (Figure 11a) has a lot of plate-shaped crystals of calcium hydroxide (C-H), needle-shaped calcium silicate hydrate (C-S-H), and other crystals such as ettringite (AFt) [57]. Figure 11b shows that C-S-H is less common and is loose, and the crystals are larger than the other two samples, which is consistent with the XRD result. Figure 11c shows many needle-shaped crystals (C-S-H) stuck to the crumb rubber tightly, consistent with the results of Chou et al. [33] and Liu [35].

#### 3.3.4. Contact Angles

In general, most cement–matrix materials are hydrophilic [58,59]. However, the hardened cement paste developed in this study has a strong hydrophobic property. A Sindatek Model 100SB contact angle goniometer was used to measure the contact angle of water droplets on the hardened cement paste. Figure 12 shows the images of water droplets on the rubber films and hardened cement pastes. Figure 12a,b shows the contact angles of the as-received rubber and the treated rubber. The contact angle of the treated rubber is less than that of the as-received rubber. It proves that the treated rubber is more hydrophilic than the as-received rubber. It is due to the production of polar functional groups on the rubber surface. Figure 12c,e,g shows contact angles of cement pastes at the initial stage. The contact angle on the pure cement paste is between 28° and 29°, and that on the cement paste with the as-received crumb rubber is 22~23°. However, the paste with the treated crumb rubber has a contact angle between 77° and 78°. The samples were kept in ambient conditions (23 °C, 75% humidity) for 10 min. The results are shown in Figure 12d,f,h. It is clear that the contact angle of the paste with the treated crumb rubber is much greater than that of the other paste. In addition, the loss of water on the paste with the treated crumb rubber is much less than that on the other paste. The water on the paste with the as-received crumb rubber disappeared completely. As the samples were kept in ambient conditions, some of the water evaporated and most of the water penetrated into the paste. It is clear that for the paste with the treated crumb rubber, the water penetration is much lower than that for the other cement pastes.

A possible partial oxidation mechanism shown in Figure 5 can be used to explain the increase in the hydrophobicity of the paste with the treated crumb rubber, as the crumb rubber undergoes a series of oxidation and cracking reactions, as shown in Figure 3. Yang [44] found that most pyrolysis oils were sulfoxides/sulfones under this condition. The sulfoxides served as surfactants with double hydrophobic tails and a hydrophilic head. The hydrophobic tail was squeezed onto the surface of the paste and enhanced its hydrophobicity. Liu et al. [35] proved that sulfoxides could slow the hydration reaction rate and lead to better crystallization. With the increasing hydrophobicity of the surface and the better crystallization of the hydrates, the paste with the treated crumb rubber can enhance the hydrophobicity of the paste and reduce the penetration of water into the paste.

A possible mechanism of the hydrophobic surface of the cement paste is shown in Figure 13. In addition to the hydrophilic functional groups produced on the rubber surface, a small amount of pyrolysis oil attaches to the rubber surface. As discussed above, most of the pyrolysis oils were sulfoxides or sulfones. They served as surfactants with a hydrophobic tail and a hydrophilic head. During the hydration process, the hydrophilic end forms a strong bond with the cement, which facilitates the connection of rubber and cement hydrate. On the surface of the paste, the hydrophobic ends face the surface and create a hydrophobic surface.

#### 3.3.5. Compressive Strength

The compressive strengths of the various hardened cement pastes are shown in Figure 14. It is clear that with the addition of the as-received rubber, the compressive strength of the hardened cement paste reduces by about 45% in comparison with that of the hardened paste without rubber (56 days). It is consistent with the results in references [60,61,62]. However, the strength of the hardened cement paste with the treated crumb rubber reduced by only 3%. These experimental results prove that partial oxidization of rubber is a feasible method to improve the properties of rubcrete.

## 4. Conclusions

A hydrophobic rubberized cement paste was developed with crumb rubber as an additive in this study. The crumb rubber was pretreated with a partial oxidation reaction. Experimental results showed that the rubberized paste with the partially oxidized crumb rubber had an excellent hydrophobic property and showed an insignificant reduction in compressive strength. The following reasons could explain the improvement in the cement–matrix composite:The partially oxidized crumb rubber had more hydrophilic functional groups. These functional groups could enhance the hydration of the cement and lead to the increase in the mechanical properties of the rubberized paste.The pyrolysis oil that was absorbed on the treated crumb rubber had sulfoxide functional groups. These functional groups behaved as surfactants with a hydrophobic tail and a hydrophilic head. The hydrophobic tail was squeezed onto the surface of the paste and enhanced its hydrophobicity.With the increasing hydrophobicity of the surface and improved crystallization of the hydrates, the paste with the treated crumb rubber could reduce the penetration of water into the paste.

## Figures and Tables

**Figure 1 materials-14-03687-f001:**
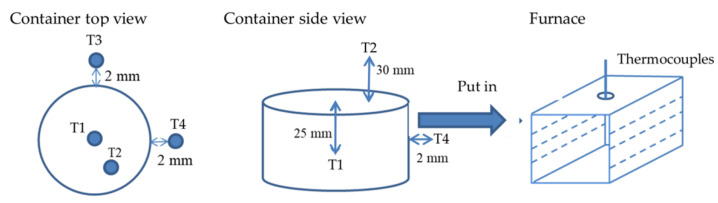
Experimental apparatus and temperature measurement points.

**Figure 2 materials-14-03687-f002:**
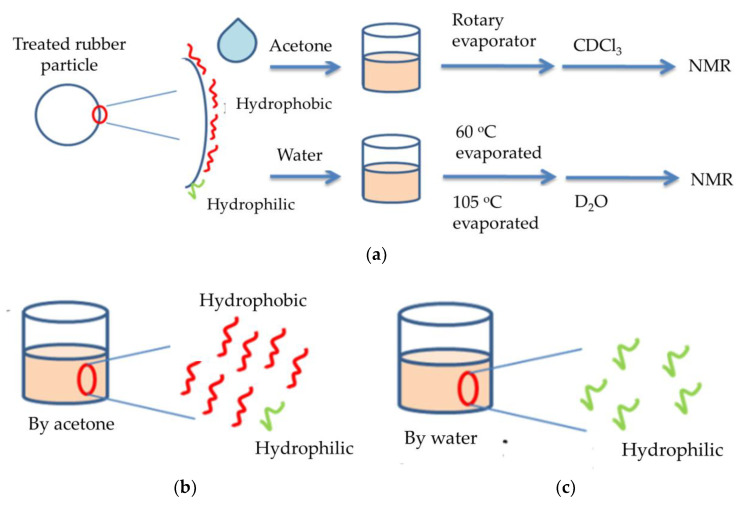
Extraction of absorbing pyrolysis oil for NMR analysis. (**a**) Extraction procedure; (**b**) components extracted with acetone; (**c**) components extracted with water. Red line: hydrophobic components; green line: hydrophilic components.

**Figure 3 materials-14-03687-f003:**
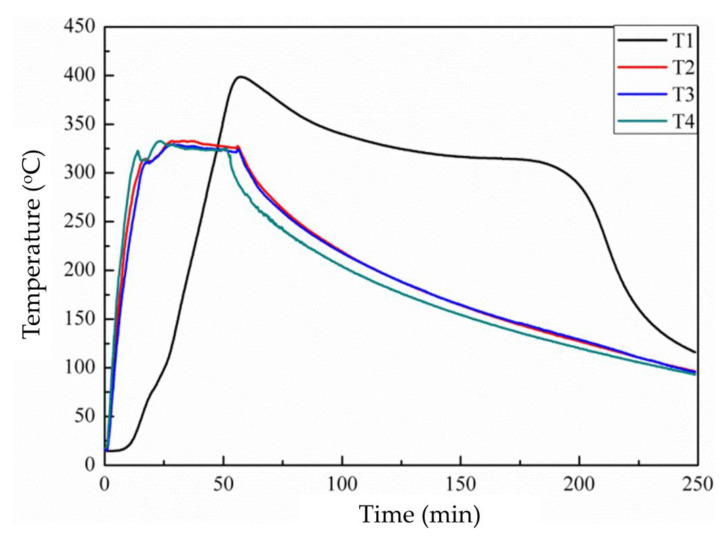
Temperature change inside and around the reactor.

**Figure 4 materials-14-03687-f004:**
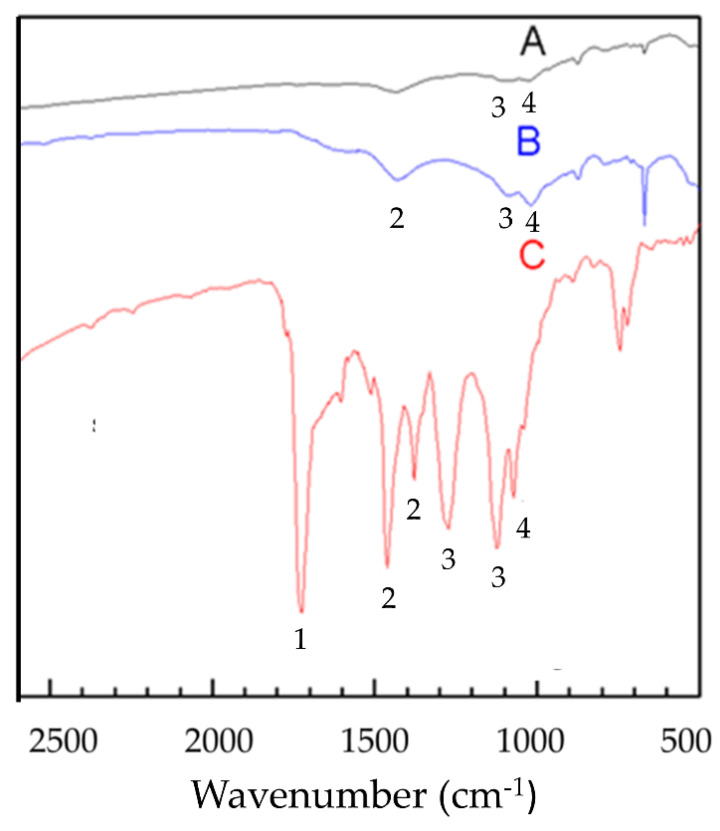
The FT-IR spectra of two kinds of crumb rubber and cracking oil. A: The as-received crumb rubber; B: the partially oxidized crumb rubber; C: cracking oil. (1: CO stretching, 2: CH bending, 3: SO_2_ stretching, 4: SO stretching).

**Figure 5 materials-14-03687-f005:**
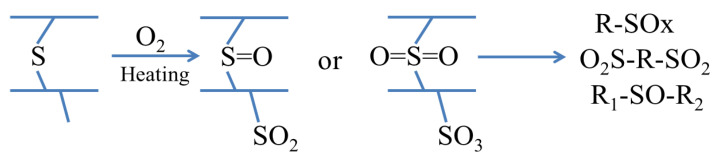
Possible mechanism of the partial oxidation and the pyrolysis reaction with limiting thermal cracking.

**Figure 6 materials-14-03687-f006:**
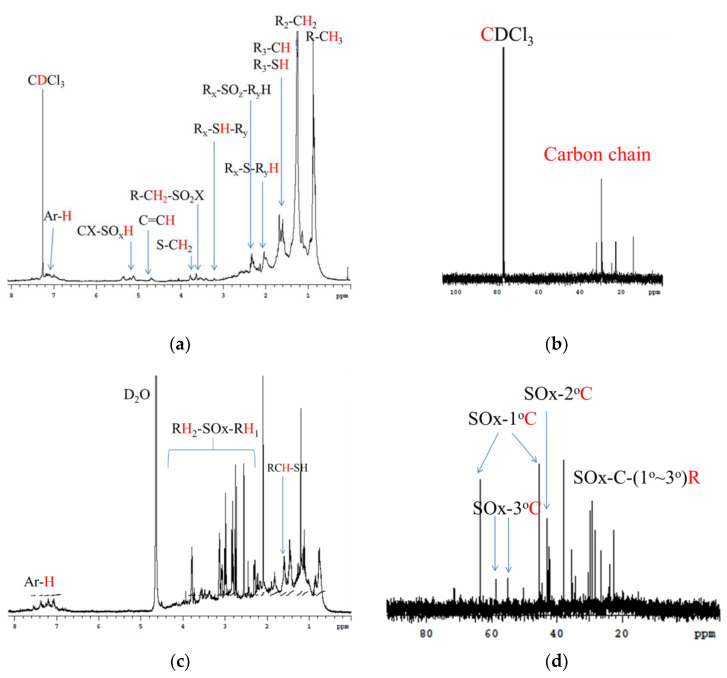
NMR spectra of cracking oil. (**a**) 1HNMR in CDCl_3_; (**b**) ^13^C NMR in CDCl_3_; (**c**) 1HNMR in D_2_O; and (**d**) ^13^C NMR in D_2_O. Samples in (**a**) and (**b**) were extracted with acetone; (**c**) and (**d**) were extracted with water; 1~3^o^ C means the degree of carbon.

**Figure 7 materials-14-03687-f007:**
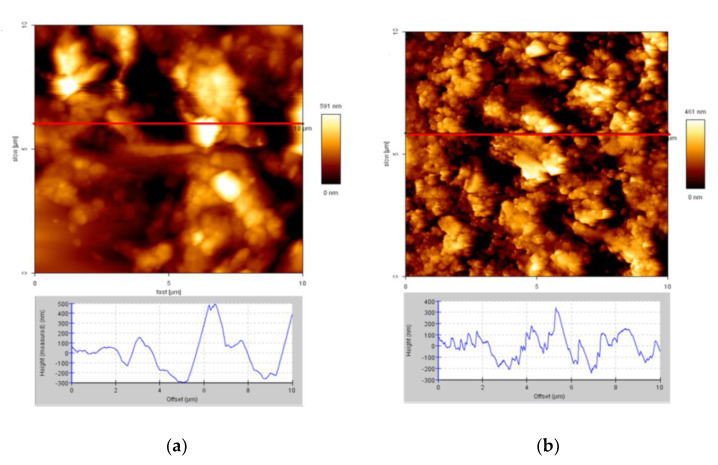
Atomic force microscopy (AFM) images (10 × 10 μm^2^) and cross-section height profiles. (**a**) As-received rubber; (**b**) treated rubber.

**Figure 8 materials-14-03687-f008:**
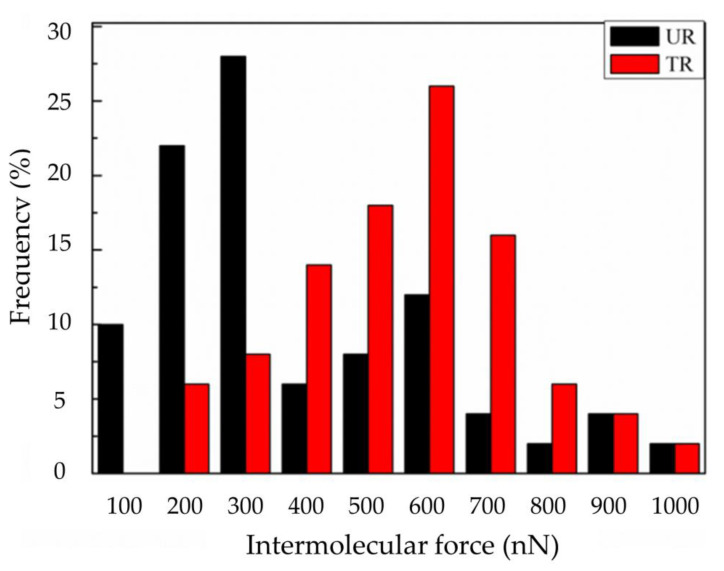
Intermolecular interaction forces between the C-S-H probe and the as-received rubber (UR) or the treated rubber (TR).

**Figure 9 materials-14-03687-f009:**
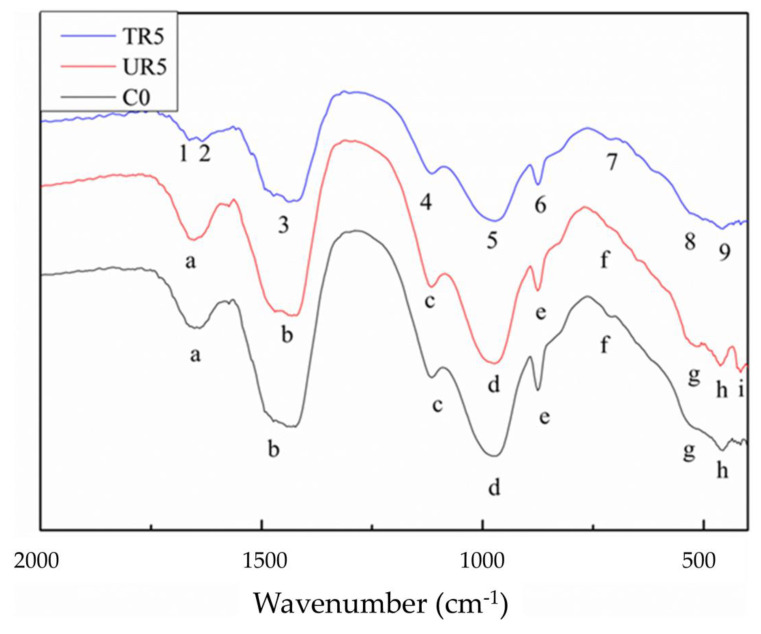
FT-R spectra of the cement pastes: C0 (black line); UR5 (red line); and TR5 (blue line).

**Figure 10 materials-14-03687-f010:**
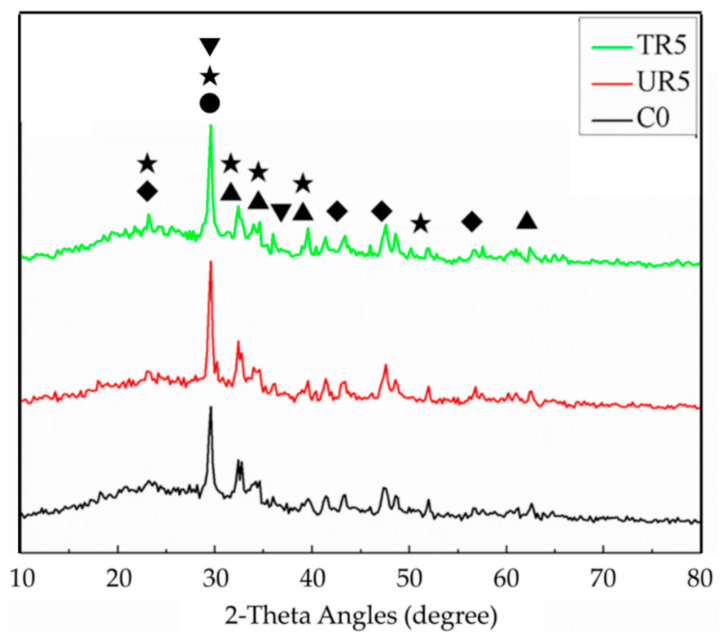
XRD patterns of the paste samples (★: C_3_S; ▲: C_2_S; ▼: CaCO_3_; ◆: C-H; ●: C-S-H.).

**Figure 11 materials-14-03687-f011:**
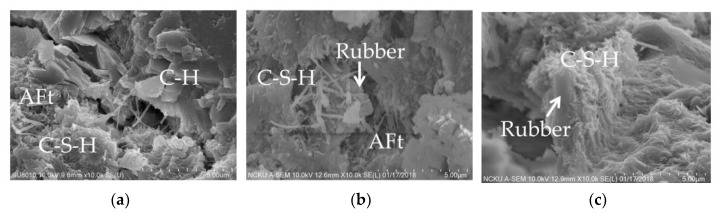
SEM images of cement pastes: (**a**) C0, (**b**) UR5, and (**c**) TR5.

**Figure 12 materials-14-03687-f012:**
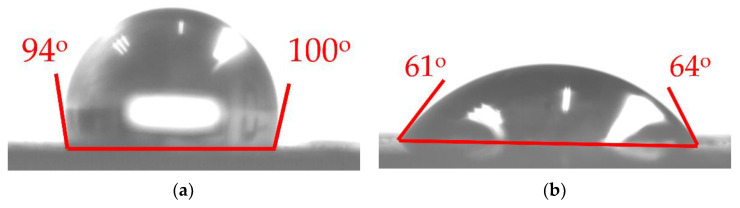
The contact angles at 0 and 10 min: (**a**) as-receive rubber; (**b**) treated rubber; (**c**) C0 at 0 min; (**d**) C0 at 10 min; (**e**) UR5 at 0 min; (**f**) UR5 at 10 min; (**g**) TR5 at 0 min; and (**h**) TR5 at 10 min.

**Figure 13 materials-14-03687-f013:**
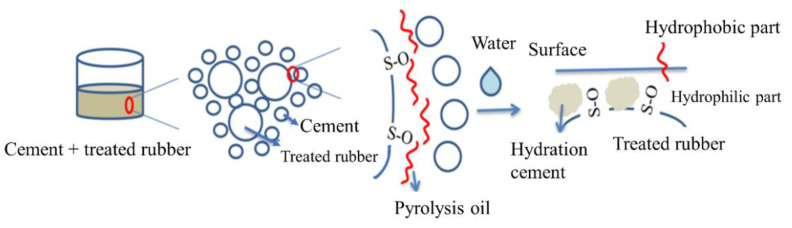
Possible mechanism of the hydrophobic cement paste. Pyrolysis oil has a long carbon chain hydrophobic part and a short carbon chain hydrophilic group.

**Figure 14 materials-14-03687-f014:**
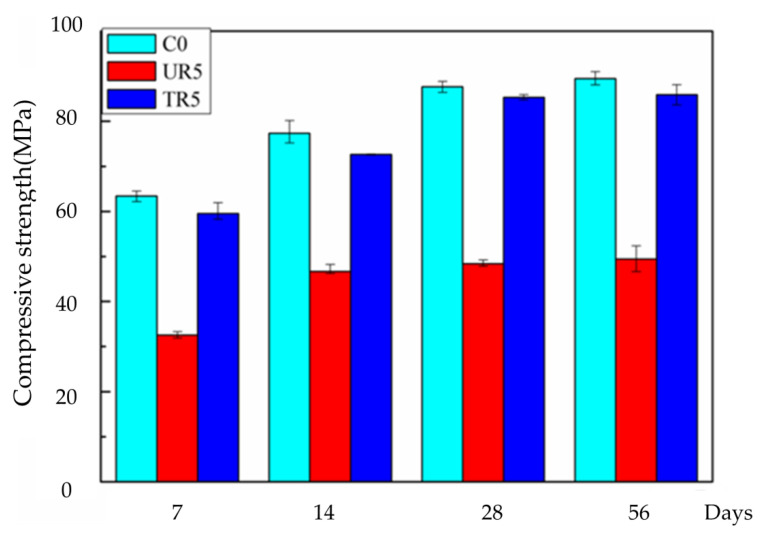
Compressive strength values of different cement pastes.

**Table 1 materials-14-03687-t001:** Energy required for cleaving sulphur or carbon bonds.

Type of Bond	Energy Required for Cleavage (kJ/mol)
C–C	348
C–S	273
S–S	227
Sx	251

**Table 2 materials-14-03687-t002:** C0, UR5, and TR5 band assignments of FT-IR.

Band	TR5 (cm^−1^)	Assignment	Band	C0 (cm^−1^)	UR5 (cm^−1^)	Assignment
1	1666	δ O–H (H_2_O)	a	1650	1650	δ O–H (H_2_O)
2	1635	δ O–H (H_2_O)				
3	1434	ν_3_ C–O (CO_3_^2^^−^)	b	1442	1431	ν_3_ C–O (CO_3_^2^^−^)
4	1114	ν_3_ S-O (SO_4_^2-^)	c	1114	1114	ν_3_ S-O (SO_4_^2-^)
5	972	ν Si–O (C-S-H)	d	972	972	ν Si–O (C-S-H)
6	875	ν_2_ C–O (CO_3_^2^^−^)	e	875	875	ν_2_ C–O (CO_3_^2^^−^)
7	713	ν C–O (CO_3_^2^^−^)	f	705	713	ν C–O (CO_3_^2^^−^)
8	532	δ Si–O–Si//Si–	g	532	509	δ Si–O–Si//Si–
9	459	δ Si–O (SiO_4_)	h	459	462	δ Si–O (SiO_4_)
			i		416	δ Si–O (SiO_4_)

**Table 3 materials-14-03687-t003:** The FWHMs of the C0, TR5, and UR5.

Sample Name	FWHMs (β)
C0	0.39921
TR5	0.42683
UR5	0.38593

## Data Availability

The data presented in this study are available on request from the corresponding author.

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
