# Peer review of "On Developing a Hydrophobic Rubberized Cement Paste"

_materials, 2021, doi:10.3390/ma14133687_

Round 1

Reviewer 1 Report

This is a nice work, and quite well presented. Apart from some concern I have regarding the presentation of the XRD results, I only have some minor comments:

  1. Line 105. "electricity supplier". I believe that the authors mean a power regulator. Otherwise please refer to it as "with controllable temperature".
  2. Line 123. What FTIR mode was used (transmittance, reflectance, atr)? Method (KBr pellets, films etc)?
  3. Line 143: Please replace was with were (it is referring to 5 microliters)
  4. Fig. 4 Assuming the measurements were performed in transmittance mode, both the spectra of the rubbers are very narrow, which is quite common in the case where a too small quantity of the material is used for making KBr pellets, or, the film was very thick to be accurately penetrated. I propose to the authors to make the same plot, but by plotting the two rubber spectra with a different y-axis than the oil, so to be easy to see all the spectral details. Finally, avoid using % in the y-axis, as you don't show any ticks. Its better to leave it just as "Transmittance" or "Transmittance (a.u.)"
  5. Line 228: Better "The results show"
  6. Line 257. Please, provide the pdf files which were used for the phase identification of the XRD peaks
  7. The authors provide the Scherrer equation and after that they just say about "largest" and "smallest" FWHMs. Please, provide the numbers, it is difficult to see it just by the presented patterns.
  8. Line 268, Fig. 10. The authors do not provide the exact phase of CaCo3. Assuming that it is calcite, its most intense peak is at 29.4° and not where the authors place it. Please, check the attribution of the peaks and provide the necessary PDF identification files you used.
  9. References. Please provide either DOI or links and not both of them

Reviewer 2 Report

The following comments and questions are made to the authors:

Style, wording, format

  • please consider the proper usage of the words: inclusion and additive, they are not always interchangeable (“addition of additives” is suggested to be avoided)
  • please revise the text, in some cases the reader needs to deal with cumbersome wording. For example raw 42, 43, 52 etc.
  • “Fourier-transfer” – mistyping: Fourier-transform
  • unlock abbreviation SBR
  • line 101: “The first type of Portland cement was used in this study.” – What do you mean by „first type”?
  • line: 105 “electricity supplier” ? – do you mean electrically controlled furnace?
  • Figure 1: Top View “T2” appears twice

Comments and questions on the content

  1. In the introduction (line 37) the authors state the following: „The addition of these additives can improve the properties of the composites, such as strength, stiffness, fatigue, toughness, etc. However, the hydrophobic problem is seldom mentioned.”
    This statement can be discussed, since a number of recent papers treat the problem of hydrophobicity of cement composites, such as cement with added graphene nanoplate, silicone hydrophobic powder, nano SiO2 or treated mica powder…
    For example:
    Multifunctional cementitious composites with integrated self-sensing and hydrophobic capacities toward smart structural health monitoring By: Dong, Wenkui; Li, Wengui; Zhu, Xinqun; et al. CEMENT & CONCRETE COMPOSITES Volume: ‏ 118     Article Number: 103962   Published: ‏ APR 2021

or
Influence of a novel hydrophobic agent on freeze-thaw resistance and microstructure of concrete, By: Zhang, Bo; Li, Qingbin; Niu, Xujing; et al. CONSTRUCTION AND BUILDING MATERIALS   Volume: ‏ 269     Article Number: 121294   Published: ‏ FEB 1 2021

  1. Please give a more detailed description of the specimens preparation: how the 300-600 micrometre rubber sheets were prepared, how the cement samples were made, what was the w/c ratio etc.?
  2. Please give more details on the cement test specimens used for the various technique measurements. How they were prepared, what sizes, quantities were used (if adequate), at what level of hydration the measurements were made? For example were all the different testing measurements - XRD, SEM, compression strength etc. - made on 28 days samples? If not, how the results can be compared?
  3. A more clear distinction between the hydrophobicity of the cement specimens surface (nicely shown by the contact angle measurements) and the hydrophobicity (or in some cases the hydrophilic character) of the rubber sheets or pyrolysis oil would make the text more easily understandable.
  4. In this study the rubber was composed of natural rubber and SBR. The two kinds of rubbers have distinct oxidative characteristics. The oxidative degradation of the SBR is mainly crosslinking and hardening, while for the natural rubber the chain scission and softening is characteristic. The authors do not make the distinction between the two different types of rubbers. Why?
  5. The thermal oxidatioin conversion can reach up to 98% level at 200°C for a 12 minutes heating time [Guo, L., Huang, G., Zheng, J. et al. Thermal oxidative degradation of styrene-butadiene rubber (SBR) studied by 2D correlation analysis and kinetic analysis. J Therm Anal Calorim 115, 647–657 (2014). https://doi.org/10.1007/s10973-013-3348-0]. How this is related to the 320°C and 50 minutes treatment used by the authors? Of course the conditions are important. Could you comment on this?
  6. Could the Authors provide quantitative FWHM values for the C-S-H peaks from the XRD spectra? The graph itself is not an adequate proof of the statements about the crystallites sizes.
  7. On Figure 4. the peaks of the A and B curves cannot be identified. Please make a new graph, similar to the Figure 9. and provide a more detailed comparison of the 3 curves.
  8. The hydrophobic character of the cement/concrete composite is an advantageous property of the building material. However the question arises: isn’t the hydrophobic character perverting the completion of the hydration? What is its influence on the hydration?

10. The compression strength measurements were made at 56 days instead of the usual 28 days. Why? The question arises: what would the 28 days compression strength measurements show? May it happen that they would prove a slowed down hydration process? Wouldn’t that be a disadvantage?

Round 2

Reviewer 2 Report

  • The details of the samples preparation are important for the readers, especially if they would like to further use the results of this article. For example the w/c=0.35 ratio should be mentioned in the article.
  • Instead of "first type cement" the correct usage is "type I cement"

Author Response

Compositions of tested pastes and details of their preparation should be described in the part “Materials and Methods”. Water to cement ratio and curing conditions of the pastes should be given.

We have added the description about the conditions of the specimens (Line 153 to 155).

The compressive specimen (50 x 50 x 50 mm) was prepared with a w/c value of 0.35. There were six specimens for each curing age. The curing process was carried out in saturated lime water, and the temperature was controlled at 23±2°C.

In two places in this work there is: “Fourier-transfer”. It should be rather “Fourier-transform”. Please check it.
We have replaced the word “transfer” with “transform” (Line14 and 74).

In lines 245, 246, there is: “(band a, 1666, 1635)”. It should be: “(band a, 1666 cm-1, 1635 cm-1)”.
We have added the cm-1 behind the two number (Line 252).

In lines 261, 262, there is: “Figure 10 and Table 3 show that the FWHM of C0 is the largest, and that of UR5 is the smallest.” Please, check if the statement is consistent with the data presented in Table 3.
We have corrected the sentence “TR5 is the largest” (Line 272).

In two places there is: “The hydrophobic tail was excluded towards the surface of the paste and enhanced its hydrophobicity.” Is the word “excluded” properly used in this case?
We have corrected the sentence “The hydrophobic tail was excluded towards the surface of the paste and enhanced its hydrophobicity.” to “The hydrophobic tail was squeezed onto the surface of the paste and enhanced its hydrophobicity.” (Line 316 and 355).

Please check the correctness of description of References, e.g. in the case of the reference No. 34 the title of the article is different than the one found on the given link. Moreover journal name was not given. In line 400 there is: “nano-sio2”, it should be: “nano-SiO2”. In line 438 there is: “(gtr)”, it should be “(GTR)”. In line 458 there is: “c–h bonds”, it should be: “C–H bonds”. Also check other references in this regard.

We have corrected these wrong places. The reference 34 is the thesis and correct the format. The other wrong places are corrected (Line 409, 447 and 467).